Accounting for spatial habitat and management boundaries when estimating forest bird population distribution and density: inferences from a soap film smoother

http://orcid.org/0000-0001-7008-923X Camp Richard J. 1 2 rcamp@usgs.gov
http://orcid.org/0000-0002-9640-6755 Miller David L. 1 3
Buckland Stephen T. 1
Kendall Steve J. 4
1 School of Mathematics and Statistics, Centre for Research into Ecological and Environmental Modelling, University of St Andrews , St Andrews , United Kingdom
2 United States Geological Survey, Pacific Island Ecosystems Research Center , Hawai‘i National Park, Hawai‘i , United States
3 Current Affiliation: Biomathematics and Statistics Scotland, Dundee, Scotland and UK Centre for Ecology & Hydrology (UKCEH) , Lancaster Environment Centre, Lancaster , United Kingdom
4 U.S. Fish and Wildlife Service , Hilo, Hawai‘i , United States
Wang Qiang
Electronic publication date: 2023 Jun 14
Publication date: 2023
Volume: 11
Electronic Location ID: e15558
Received 2022 Nov 21; Accepted 2023 May 23
Copyright year: 2023
License: This is an open access article, free of all copyright, made available under the Creative Commons Public Domain Dedication. This work may be freely reproduced, distributed, transmitted, modified, built upon, or otherwise used by anyone for any lawful purpose.
License URL: https://creativecommons.org/publicdomain/zero/1.0/

Keywords: Density estimation, Loxops coccineus, Point-transect distance sampling, Soap film smoother, Spatial modelling

Funding: Centre for Research into Ecological & Environmental Modelling University of St Andrews U.S. Geological Survey Funding for Richard J. Camp was provided through a studentship from Centre for Research into Ecological & Environmental Modelling, University of St Andrews and by the U.S. Geological Survey. There was no additional external funding received for this study. The funders had no role in study design, data collection and analysis, decision to publish, or preparation of the manuscript.

==============================
Birds are often obligate to specific habitats which can result in study areas with complex boundaries due to sudden changes in vegetation or other features. This can result in study areas with concave arcs or that include holes of unsuitable habitat such as lakes or agricultural fields. Spatial models used to produce species’ distribution and density estimates need to respect such boundaries to make informed decisions for species conservation and management. The soap film smoother is one model for complex study regions which controls the boundary behaviour, ensuring realistic values at the edges of the region. We apply the soap film smoother to account for boundary effects and compare it with thin plate regression spline (TPRS) smooth and design-based conventional distance sampling methods to produce abundance estimates from point-transect distance sampling collected data on Hawai‘i ‘Ākepa Loxops coccineus in the Hakalau Forest Unit of the Big Island National Wildlife Refuge Complex, Hawai‘i Island, USA. The soap film smoother predicted zero or near zero densities in the northern part of the domain and two hotspots (in the southern and central parts of the domain). Along the boundary the soap film model predicted relatively high densities where ‘Ākepa occur in the adjacent forest and near zero elsewhere. The design-based and soap film abundance estimates were nearly identical. The width of the soap film confidence interval was 16.5% and 0.8% wider than the width of the TPRS smooth and design-based confidence intervals, respectively. The peaks in predicted densities along the boundary indicates leakage by the TPRS smooth. We provide a discussion of the statistical methods, biological findings and management implications of applying soap film smoothers to estimate forest bird population status.

Introduction

Understanding species’ distribution, density and abundance are cornerstones of ecology, conservation and species management. Distance sampling is a common method for estimating densities, accounting for imperfect detection where a set of points are visited and the distance to each detected bird recorded (point-transect distance sampling (PTDS); Buckland et al., 2015). Recent developments in spatial modelling take advantage of spatially referenced data allowing for the modelling of density as a function of environmental covariates (e.g., spatial location, habitat type and elevation; Miller et al., 2013), and can be used to create maps of species density, which are useful for management. Camp et al. (2020) modelled the spatio-temporal densities of Hawai‘i ‘Ākepa (Loxops coccineus; hereafter ‘Ākepa) across a 31-year time series using a two-stage model-based smoothing method. This method captured spatial and temporal correlation in bird densities, thus reducing estimates of uncertainty. However, the penalized thin plate regression spline-based smoother used by Camp et al. (2020) predicted non-zero densities across the forest boundary into unsuitable habitat, a modelling artefact commonly termed leakage (Wood, Bravington & Hedley, 2008). The leakage arose because the topological structure and contours extend across the boundary (Fig. 1).

Figure 1 (A) Location of the study area (black diamond) within the Hawaiian Islands. (B) Study area showing survey points in the open-forest (black polygon), closed-forest (green polygon), and reforested pasture stratum (orange dots in orange polygon) sampled during the 2002 forest bird survey at the Hakalau Forest Unit of the Big Island National Wildlife Refuge Complex, Hawai‘i Island.

Open dots are survey points without detections and black dots are points with Hawai‘i ‘Ākepa Loxops coccineus detections. Blue polygon represents the expanded forest study area. (C) Plot of the study area (black polygon) and survey points (black dots) with contours (thin lines) show the estimates of the smooth from a thin-plate regression spline fitted GAM, estimates are on the linear predictor scale. The predicted densities outside the forest stratum boundary were modelling artefacts, commonly termed “leakage.” Predictions were made over a larger area to illustrate that the TPRS model suffers from leakage, particularly along the western boundary.

Conventional distance sampling combines model- and design-based methods. Model-based methods are used to estimate detection probability, allowing plot counts to be adjusted for undetected animals, and estimated plot densities are extrapolated using design-based methods to estimate abundance in the survey region (Buckland et al., 2015). We can substitute the design-based method with a spatial model, giving a completely model-based solution (Fig. 2). Hedley & Buckland (2004) developed an approach in which animal counts in transect segments were modelled using generalized additive models (GAMs; Wood, 2017). Detectability was first estimated for conventional distance sampling, and this entered the GAM as an estimated offset, which adjusts counts for undetected animals.

Figure 2 Conceptual data flowchart of the modeling framework to estimate densities and 95% confidence intervals (CI) using distance sampling (stage 1) and generalized additive model (GAM; stage 2) methods.

Detection functions can be fitted using conventional distance sampling (CDS), or covariates can be included in the detection function model, using the multiple covariate distance sampling engine (MCDS) to estimate probability of detection. In the second stage, the fitted detection function model is either used to estimate densities with 95% CIs using design-based methods, or estimated detection probabilities are included in the GAM as an offset, so that counts are adjusted for undetected animals. Rounded rectangles indicate data entering the process flowchart that pass through operations, rectangles. Arrows direct the sequence in processes, and output is represented by a rectangle with wavy base that is either passed from the first to second stage, the offset, or as estimated densities.

‘Ākepa are an internationally and federally endangered Hawaiian honeycreeper (Fringillidae; Pratt, 1994, BirdLife International, 2016) endemic to Hawai‘i Island, USA. ‘Ākepa are routinely monitored at the Big Island Unit of the Hakalau Forest National Wildlife Refuge Complex (hereafter Hakalau) where the species occurs in native-dominated montane forest. Afforestation of pasture adjacent to forest habitat is a primary management objective at Hakalau (USFWS, U.S. Fish & Wildlife Service, 2010). Leakage is a concern in the spatial analyses because until recently no ‘Ākepa were detected in the pasture stratum (Camp et al., 2016, Paxton et al., 2018) and the afforestation in the pasture had not progressed to produce suitable ‘Ākepa habitat. While densities in the pasture stratum could easily be excluded by considering only density estimates from the forest strata, the leakage draws into question the applicability of the model. That is, do ‘Ākepa occur at relatively high densities right to the pasture-forest boundary or do densities taper prior to declining rapidly to zero beyond the stratum edge? This question can be addressed by applying a soap film smoother to the data (Wood, Bravington & Hedley, 2008).

Two important considerations in spatial data analysis are the structure of the data and the domain shape. The data used here are from a single survey of ‘Ākepa data from Hakalau conducted in 2002, and we include only spatial location (easting and northing) as a covariate in the spatial model. Sampling coverage was relatively uniform over the forest stratum thus minimizing extrapolation when making predictions. Managers and policy makers require population estimates at multiple spatial scales such as the species’ distribution, the distribution of suitable habitat or representative land-use, and political or management boundaries. As management plans develop there is increasing mismatch between the sampling extent or resolution and areas of interest, requiring extrapolation to unsampled areas. This can also occur with changes in political boundaries through land acquisition or designation for conservation (USFWS, U.S. Fish & Wildlife Service, 2010). Here, we expanded the forest stratum study area addressed in Camp et al. (2020) to include the surveyed portions of the open- and closed-forest strata, and extended the extrapolation area to coincide with plausible management units (Fig. 1).

In this article we model the 2002 ‘Ākepa forest stratum data, selecting the detection function, smoother and response distribution to estimate a density surface, and estimate population abundance and uncertainty. We compare the soap film smoother estimates to those of a TPRS smooth and design-based analysis. Finally, we identify how the soap film smoother handles extrapolation beyond the range of the data.

Methods

Study area

As previously described in Camp et al. (2020), Hakalau (19°51′N, 155°18′W) was established in 1985 and is actively managed to preserve native forest birds, rainforest plants and their habitats. The 15,390-ha montane forest is dominated by native ‘ōhi’a lehua Metrosideros polymorpha and koa Acacia koa with a mixture of native and non-native understory plants. Temperature averages 15 °C with annual variation <5 °C, and annual precipitation averages 2,500 mm with a maximum of 6,100 mm (Juvik & Juvik, 1998). Hakalau is roughly divided into three strata: afforested-pasture, open- and closed-forest (Fig. 1; see Camp et al., 2016 for a detailed map). The afforested-pasture stratum extends from the western edge of the open-forest stratum to the western edge of the Hakalau boundary, covering an area of 1,271 ha, with elevation between 1,650 and 2,000 m above sea level (asl). The afforested-pasture was intensely grazed through the mid-1980s and is being gradually reforested through outplanting koa. The open-forest stratum occurs at an elevation between 1,400 and 1,920 m asl and encompasses an area of 3,061 ha. Previously heavily grazed, the open-forest stratum has undergone natural regeneration since the removal of cattle in 1988 (Maxfield, 1998). The closed-forest stratum extends down slope from the eastern edge of the open-canopy stratum to the eastern edge of the Hakalau boundary, with elevation between 1,400 and 1,700 m asl. The closed-forest stratum was least modified by grazing and surveys commenced in 1999 covering an area of 1,541 ha.

Bird surveys

Following the creation of Hakalau, in 1987 refuge staff initiated an annual abundance monitoring program by establishing 350 point samplers (hereafter, points) on 14 transect lines following a systematic random design spanning the upper elevations of Hakalau. U.S. Fish and Wildlife, Big Island National Wildlife Refuge Complex, coordinated and conducted forest bird monitoring on the Hakalau Forest Unit. The distance among points is approximately 150 m with transects being either roughly 500 or 1,000 m apart (Fig. 3). For this analysis we selected a single survey from the ‘Ākepa time series based on broad sampling of the study area and sufficient numbers of detections. In 2002, 289 points were sampled using PTDS methods within the forest stratum where 276 ‘Ākepa were detected at 121 points. Observers recorded the species, detection type (heard, seen, or both), and horizontal distance from the station center point to individual birds detected during an 8-min count. Observers also recorded cloud cover, rain intensity, wind strength on the Beaufort scale, and gust strength on the Beaufort scale.

Figure 3 Plot of forest stratum study area (red polygon) with points and knots located within the soap boundary (blue polygon).

Open circles are sampling points without detections and green dots are points with detections (scaled by numbers of detections; range 1–6). Knots within the boundary are orange circles.

Study boundary

To exemplify extrapolating to management unit or political boundaries we set the western edge of the boundary to the coordinates of the pasture-forest boundary, while the north, east and south boundaries were squared off (hereafter, referred to as the “soap boundary”) using the function locator (Becker, Chambers & Wilks, 1988) (Fig. 3). In addition to defining the soap boundary, soap film smoothing also required a priori specification of the number and location of knots inside the boundary. We defined a set of knots on a regular grid with locations every 730 m east and 670 m north across the study area (Fig. 3).

Stage 1 modelling detectability

Using the R package Distance (Miller, 2017, Miller et al., 2019) we fitted half-normal and hazard-rate key detection functions without either adjustment terms or covariates (Fig. 2). Preliminary analysis revealed key detection functions with adjustment terms were not strictly monotonic, and there were insufficient detections per factor to model covariates. Data were truncated at a distance w where the estimated detection probability (using a preliminary detection function model) was about 0.1. Model selection for the detection function used AIC (Burnham & Anderson, 2002) and model fit was evaluated with a Cramér-von Mises test (Buckland et al., 2015).

Stage 2 design-based density estimation

For comparative purposes we estimated the ‘Ākepa abundance using conventional distance sampling methods (Fig. 2; Buckland et al., 2015). We combined the estimated detection probability with counts to estimate densities as birds ha−1 using package Distance. Abundance was estimated as density times the area inside the boundary (5,671.8 ha). Variance and 95% CI were calculated using analytic methods (Buckland et al., 2015).

Stage 2 model-based density estimation

Soap film smoother model

To control leakage when building spatial models of animal abundance (Fig. 2), Miller et al. (2013) suggest using a soap film smooth (Wood, Bravington & Hedley, 2008) to fit the surface instead of a thin plate regression spline (TPRS) smoother (Wood, 2003). The soap smoother consists of two component smooths: one “boundary smooth” ( fΩ) modelling the values around the edge of the domain (which we refer to as Ω) and “interior smooth” ( fx,y) modelling observations inside the domain (the space inside Ω). Both smooths were estimated from the data with a cyclic spline smoother used along the boundary. This setup allows the boundary condition to be enforced while allowing smooth departures into the interior of the domain (Wood, 2017, pg 223–227).

The model was fitted in R using the mgcv package (Wood, 2021). The soap model has the form:

log⁡{E(nk)}=fΩ(Eastingk,Northingk)+fx,y(Eastingk,Northingk)+log⁡(ν^),

where nk is the bird count at the k-th point, fΩ is the boundary smooth that was estimated and defined with a maximum basis size of 20. fx,y is the interior smooth that was fit with 108 interior knots (equivalently a maximum basis size of 108). Although the maximum number of basis dimension is arbitrary, it was sufficiently large to permit enough degrees of freedom to represent the underlying smooth reasonably well. ν^ is the effective area searched (estimated from the detection function), and the offset is the log of ν^. Restricted maximum likelihood (REML) was used to estimate model parameters, and density estimates were predicted from the fitted model to generate densities and standard errors.

Since the response variable, nk, is a count, we evaluated three possible response distributions: Poisson, negative binomial and Tweedie, all with a log link function. Based on preliminary analyses the soap model was fitted with a negative binomial response distribution with a log link (see Supporting Online Information Appendix A for response distribution modelling, evaluation and selection). We refitted the soap model, with intercept, with the deviance residuals as response to determine if any pattern remained in the residuals following methods by Marra, Miller & Zanin (2012) and Wood (2017). If there is no remaining pattern in the residuals, the refitted model effective degrees of freedom (EDF) should be near zero for each term (residuals are randomly distributed about a plane). Model assumptions were checked and results are presented in Supporting Online Information Appendix A.

TPRS smoother model

We fitted a spatial, penalized spline-based (TPRS) smoother of the form

log⁡{E(nk)}=f1(Eastingk,Northingk)+log⁡(ν^)

where f1 is a two-dimensional TPRS smooth function. The other variables are as above. The basis was set to match the maximum complexity of the soap smoother (above), with 127 basis dimensions, that when smoothed yielded a total of 126 knots (one degree of freedom is allotted for the identifiability constraints; Wood 2016 help pages). As with the soap model, REML methods were used for estimating smoothing parameters and a negative binomial response distribution with a log link was used to model the response distribution. Density surface model and corresponding standard errors were generated from the fitted model.

Comparison between estimates

Estimated coefficients cannot be directly compared between most spline models. Instead, we predicted density and uncertainty estimates and mapped the distribution of ‘Ākepa densities within the soap boundary on a grid with cells 200 m × 200 m. We also evaluated the model predictions at even distances along the boundary. Estimates of abundance and 95% confidence intervals (CI) were computed from the posterior distribution of the fitted soap model following methods described by Wood (2017) and detailed in Camp et al. (2020). A total of 10,000 replicate parameter value sets were drawn from the posterior distribution of the model coefficients, β^, assuming a multivariate normal distribution. We estimated abundance for each of these parameter samples and calculated the mean of the replicate sets, SE as the standard deviation of the replicates, and 95% CIs were computed from the 2.5th and 97.5th quantiles. The detection function uncertainty was not propagated with the uncertainty from either model-based approach as the detection probability does not contain any spatial variability (modelled without covariates since there were no detection covariates). All analyses were conducted using the R statistical language and code is provided in Appendix B. We computed the change in uncertainty as the percentage change in interval widths (CIW) between soap and TPRS, and soap and Distance abundance estimates expressed as (CIWTPRSCIWsoap−1)×100% and (CIWDistanceCIWsoap−1)×100%, respectively.

Results

Design-based density estimation

Using a preliminary detection function model, truncation was set at 58 m. The AIC for the hazard-rate detection function model was 11 units smaller than that of the half-normal model (Table 1). The Cramér-von Mises test was non-significant at the α=0.05 level indicating that the detection function provided a satisfactory fit to the distance histogram (Table 1). Inspection of diagnostic plots also indicated that the model adequately fit the data (Appendix C). The shoulder of the detection function extends out to 30 m before decaying rapidly. The estimated detection probability of a bird that was within 58 m of a point was 0.631 (SE = 0.035) and the effective area surveyed per point was 6,668.6 m2.

Table 1 Detection function model selection statistics and parameter estimates.

Key function without adjustment terms ranked by AIC. Presented are the model unweighted Cramér-von Mises (C-vM) statistic and p-value, and the estimated detection probability with standard error.

Key function	Δ AIC	C-vM	p-value	Pa^	se ( Pa^)	
Hazard-rate	0	0.042	0.920	0.631	0.035	
Half-normal	11.094	0.428	0.061	0.443	0.040	

Soap film smoother

Our soap film smoother explained 54.6% of the deviance in the data using 2.98 of 18 boundary degrees of freedom and 14.45 of 108 interior degrees of freedom, and the index to ensure adequate complexity was 1.1 (desired value should be near one for each term). The negative binomial dispersion parameter was 10.55 indicating that the counts were over-dispersed.

The main advantage of using the soap smooth is that there was no leakage across the pasture-forest boundary (Fig. 4, left panel). The density surface maps showed an ‘Ākepa density hotspot in the southern portion of the domain that extended north-east to a second hotspot in the central portion of the domain (Fig. 4, left panel). Both soap smooth hotspots occurred interior to the boundary, even though the parametrization allowed for fitted values to persist adjacent to and including the boundary. Densities throughout the northern portion of the domain were zero or near zero. The uncertainty estimates portrayed the same pattern with large SEs predicted in the southern portion that extended north-east to the central portion of the domain (Fig. 4, right panel). SEs in the northern portion of the domain were near zero while SEs were moderate adjacent to the south boundary and throughout a swath along the south-eastern boundary.

Figure 4 Predicted density surface map of Hawai‘i ‘Ākepa Loxops coccineus densities (birds ha−1; left column) and SE (right column) for the 2002 dataset using a soap film (top row) and TPRS (bottom row) based smooths.

Fitted values projected to the soap film boundary.

On the boundary the largest densities predicted by the soap smoother occurred in the south-western corner of Hakalau (labelled zero in Fig. 5) with densities of about 1.25 birds ha−1. Densities declined smoothly to near zero by and along the northern boundary (occurring between 300 and 600 in Fig. 5) and then increased progressively along the eastern boundary until again reaching maximum densities at the south-western corner.

Figure 5 (A) Predicted Hawai‘i ‘Ākepa Loxops coccineus densities (birds ha−1) along the boundary for the 2002 dataset using the soap film (red) and TPRS (blue) smoother. (B) Location of distance around the soap film boundary (point 0 and 1,000 are at the same location).

TPRS smoother

The TPRS smoother had an estimated negative binomial dispersion parameter of 12.40 and explained 55.1% of the deviance. There was a clear pattern of lower densities in the north than in the south part of Hakalau. The EDF on the Easting and Northing smoothing term was greater than zero, significant and the function was nonlinear (Supporting Online Information Appendix A Table 2; Supporting Online Information Appendix A Fig. 7).

There were two density hotspots and two SE hotspots in the density surface maps predicted from the TPRS model (Fig. 4, bottom left and right panels respectively). The hotspot in the south portion was relatively large spanning north-east and extending to the south and west boundaries. The second hotspot occurred in the central portion of the domain and extended to the east boundary. The contours of the TPRS model continued beyond the soap boundary, which resulted in very convoluted densities along the boundary (Fig. 5).

Comparison of smooths predictions

Both smoothers had visually similar cold and hotspots in the domain with similar predicted densities and SEs (Fig. 4). Both smoothers predicted zero or near zero densities in the northern part of the domain. The extent of the coldspot was much larger for the soap smooth than for the TPRS smooth extending from the central hotspot to the north boundary, while the TPRS smooth predicted approximately zero densities to about half the area of that predicted by the soap smooth. The large hotspot in the southern part of the study area was similar in shape and extent between the two smoothers. The local hotspot in the central part of the study area was more symmetrical for the soap smooth than for the TPRS smooth which was more triangularly-shaped. For the TPRS, both hotspots extended to the boundary. In contrast, both soap smooth hotspots occurred interior to the boundary, even though the parametrization allowed for fitted values to persist right to and including the boundary.

Greater differences between the two smoothers were observed in the fitted SE than the fitted density estimates (Fig. 4). The TPRS and soap both had two hotspots, but the SE in the central part of the domain was adjacent to and extended along the boundary. The equivalent soap SE hotspot was more centrally located between the east and west boundaries and the SE estimates were generally smaller. The TPRS and soap smooth global maximum SEs were in the southern part of the domain and the soap hotspot portrayed a patchwork of varying SEs while the TPRS SEs were relatively uniform across the entirety of the hotspot with less cell-to-cell variability.

The soap-film basis argument controlling the boundary smooth eliminated the extreme rough densities of the TPRS smooth (Fig. 5). On the boundary the largest densities predicted by the soap smooth occurred in the south-west corner of Hakalau (labelled zero in Fig. 5). Densities declined smoothly to near zero by and along the northern boundary (occurring between 300 and 600 in Fig. 5) and then increased progressively along the eastern boundary until again reaching maximum densities of nearly 1.25 birds ha−1 at the south-west corner. Predicted densities from the TPRS smooth were more rough with peaks at points 80, near 700 and at 950 along the boundary (Fig. 5). There were minor peaks at about points 100, 200 and 800. Each of these peaks occurred where contours intersected the boundary (compare Fig. 1, right panel, and Fig. 5). In between each peak the TPRS smooth predicted small to very small densities before increasing rapidly to the next peak. Similar to the soap smooth, the TPRS smooth predicted near zero densities along the northern boundary, between points 400 and 600. The peaks along the boundary indicates leakage by the TPRS smooth.

Abundance estimates

The soap film smooth estimate was 6,518–9,630 ‘Ākepa with a median point estimate of 7,743 ± 794 (Table 2). The TPRS smooth produced similar estimates to the soap estimates and the abundances of the two approaches differed by more than 100 birds with a TPRS median point estimate of 7,619 ± 673 birds. Estimates of the 2002 ‘Ākepa abundance using design-based, conventional distance sampling analyses were similar to the soap smooth estimates. The 95% CIs of each approach bracketed the mean abundance estimates of the other two approaches. The soap CIW was 16.5% wider than the TPRS CIW, and was 0.8% wider than the Distance CIW.

Table 2 Abundance, standard error, coefficient of variation (CV), and 95% confidence limits (LCL = lower 95% CI limit and UCL = upper 95% CI limit) estimates for the soap film and TPRS smooths and design-based methods fitted to the 2002 ‘Ākepa for the soap boundary.

Estimator	Abundance	SE	CV	LCL	UCL	
Soap	7,734	794.10	0.102	6,518	9,630	
TPRS	7,619	672.50	0.088	6,494	9,092	
Distance	7,713	782.55	0.101	6,323	9,410	

Discussion

Statistical methods

We used a soap film smoother to model spatial densities of ‘Ākepa from PTDS count data. The two dimensional soap film smoother comprises two separate but linked bases; one for the boundary and one for the film itself. In this case the soap basis arguments provide a good approach for estimating the boundary and interior surface splines. This approach of estimating densities along the boundary allowed us to answer the question posed in the Introduction: do ‘Ākepa occur at relatively high densities adjacent to the pasture-forest boundary? Densities at the boundary varied along the pasture-forest boundary as well as within the domain (see Figs. 4 and 5). As seen in Fig. 5, the predicted values on the boundary were smallest in the north-western corner and increased along the western boundary toward the south-western corner. Thus, in the southern portion of Hakalau ‘Ākepa occur at relatively high densities near to the edge of the pasture-forest boundary while in the northern portion ‘Ākepa densities were near zero inside the domain and on the boundary, as well as outside the domain in the pasture stratum.

The soap smooth model estimating the boundary reflects that ‘Ākepa densities may extend beyond the soap boundary. As seen in the ‘Ākepa densities, estimating density along the boundary is more realistic since it is not certain that the boundary value will be zero. Along the eastern and southern boundaries the coefficient indicates that the population extends beyond the boundary into the adjacent forest habitat and the boundary that poses no real physical barrier in the ‘Ākepa distribution. Fixing the boundary to zero instead of estimating it would force the smooth of the interior surface to decrease to zero at the boundary. This approach may be appropriate for island populations where birds are restricted to suitable habitat that is located within an inhospitable matrix. Examples of this include bird populations on Pacific islands such as Aguiguan in the Mariana Islands (Amidon et al., 2014), Tau in American Samoa (Judge et al., 2013) and Nihoa in the Hawaiian Islands (Gorresen et al., 2016) where suitable forest bird habitat occurs across these islands and extends to the coastal non-forest habitat or high-tide water line. For more complex systems, the approach applied here can be used to address and overcome challenges of holes (i.e., unsuitable habitat such as lakes, urban areas and agricultural fields for forest obligate species) within the domain and to model the domain boundary.

The choice of whether to start with a TPRS or soap film smoother should take into consideration features of the study area and data that may affect predicted densities. Features such as a lake or a change in habitat along the study area may result in leakage. A general approach may be to start with a TPRS smoother and if leakage is identified switch to a soap film smoother. The peaks in predicted densities along the boundary indicate leakage by the TPRS smooth, and thus the soap film smoother provides more biologically realistic ‘Ākepa densities. The most common ecological applications of soap film smoothers are in marine ecosystems, with species such as fish (Augustin et al., 2013, Polansky et al., 2018), cetaceans (Dellabianca et al., 2016) and seabirds (Grecian et al., 2016). Alternative methods that respect complex boundaries include finite element L-splines (Ramsay, 2002), geodesic low rank thin plate spline methods (Wang & Ranalli, 2007), multi-dimensionally-scaled Duchon splines (Miller & Wood, 2014), complex region spatial smoother (CReSS; Scott-Hayward et al., 2014), and barrier models (Bakka et al., 2019). Each method has several advantages and disadvantages, and comparisons conducted by Miller & Wood (2014) and Scott-Hayward et al. (2014) indicate that the CReSS and soap film methods perform better than the other methods. Moreover, soap film smoothers fit easily within the GAM framework and are readily modelled in R (R Core Team, 2017) package mgcv (Wood, 2021). Limitations of GAMs are that they are restricted to additive models, and while interactions can be captured it can be difficult to construct interactions involving many variables. The main issues with using the soap film are that it can be hard to parameterize, requiring specifying the boundary, checking model convergence, and interpreting the results. The main advantages of using GAMs are that they are easy to interpret, the flexible predictor can identify complex data patterns, and regularisation of smoothing functions avoids overfitting (Wood, 2017). Moreover, the software package mgcv (Wood, 2021) facilitates modelling smoothers including model selection and model checking, making the analyses accessible to a broad user community.

Biological findings

The hotspot in the southern part of the domain coincides with the locally abundant density estimates identified by Scott et al. (1986) at 200+ birds km--2. Similarly to the Scott et al. (1986) predictions, our results indicate that ‘Ākepa densities decrease outside the hotspot. ‘Ākepa extend outside the domain on three sides of the soap boundary but are restricted to forest habitats above 1,500 m elevation. On the northern and eastern sides of the domain this is within several 100 m of the boundaries. The ‘Ākepa distribution to the south of Hakalau continues along the 1,500 m elevation contour before terminating several kilometres south of the study area (Judge et al., 2018). The current sampling frame is centred to track the core, high density proportion of the population, and although the extent of the survey is limited, the soap smoother based density estimates appear to be a good approximation of the ‘Ākepa distribution and abundance in the region (Scott et al., 1986; Lepson & Freed, 1997).

Abundances of between 6,300 to 9,700 ‘Ākepa in the extended forest stratum boundary seems realistic for the 2002 survey in this high density ‘Ākepa population (Gorresen et al., 2009; Camp et al., 2010, 2016). Simulation studies of design-based distance sampling have shown that method to be unbiased when the critical assumptions are met or possess low bias when assumptions fail (Buckland et al., 2001, 2015). This is not necessarily the case with model-based approaches and model mis-specification can lead to bias. The standard procedure in this case is to check the residual structure of the selected model. The residual plots were qualitatively similar and the desired refitted model EDF was approximately zero for each term. While we are not able to assess bias in the soap film smoother estimates, it is insightful to compare these estimates to those produced using the design-based approach assuming that the latter approach is unbiased. In this case it appears that both the soap and TPRS smoothers are unbiased. The 95% CIs among the three estimators overlapped the abundance estimate produced by the other models (see Table 2). The coefficient of variation of abundance was very small at about 10% for the Distance, soap, and TPRS models.

Along the southern portion of the pasture-forest boundary ‘Ākepa occur in relatively large densities (see Fig. 5). The combination of suitable habitat and high densities provides the potential for ‘Ākepa to colonize the pasture-stratum naturally. It was not until 2011 that ‘Ākepa had been detected in the afforested pasture (Paxton et al., 2018). Surveys subsequent to 2012 indicate that ‘Ākepa continue to use the koa in the pasture stratum (Steve J. Kendall, 2020, personal observations). These occurrences are likely diel migrations to locate foraging sources (Lepson & Freed, 1997); however, there is no evidence of ‘Ākepa breeding outside of established and mature forested habitats. As afforestation of the pasture stratum progresses ‘Ākepa appear poised to colonize this restored forest.

Conclusions

Extending the extrapolation area to coincide with plausible management units, i.e., making predictions outside the range of the data, should be made with caution. The Distance and soap mean abundance estimates were nearly identical; differing by 21 birds. In our case, the soap estimated boundary densities near zero along the north boundary and down both the west and east boundaries in the top quarter of the domain. Within the domain in this region densities were also near zero with very little spatial variability, reflecting that no ‘Ākepa were detected in this region of Hakalau. The soap projected the near zero estimates into the unsampled north-western portions of the domain. Had there been even a few detections in the sampled area adjacent to the extended boundary the density surface would have been less smooth which would have resulted in more variable abundance estimates, as was seen along the south-eastern boundary (see Fig. 4). It is reasonable to extrapolate into the unsampled north-western portion of the domain where the forest habitat was contiguous and with inference limited to areas only in close proximity to where data were collected, and where counts, densities and uncertainty in densities were low (see Figs. 3 and 4). In the north-western corner and outside the northern boundary it is likely that ‘Ākepa were absent, or in extremely low numbers. The spatial smooth was more variable in the south-eastern corner of Hakalu (see Fig. 4, right panel) where a few ‘Ākepa were detected near the eastern ends of the transects (see Fig. 3). The model appears to have difficulty capturing the spatial correlation in this region and the reliability of extrapolating beyond the data is questionable. Extrapolation limitations did not appear to have an effect in the spatio-temporal smooth of Camp et al. (2020), likely due to borrowing information through space and time and they maintained a close proximity of survey points and the extrapolation area. Further investigation could confirm that the soap film smooth is unbiased, precise and widely applicable with forest bird survey data.

Supplemental Information

Supplemental Information 1 GAM model selection and diagnostics, R code for spatial soap film model fitted to counts, and Distance model selection and diagnostics.

Click here for additional data file.

We thank the many agency staff and volunteers who helped collect the data. All components of this study were purely descriptive, strictly non-invasive and based exclusively on passive observations. We thank Len Thomas who helped conceive the ideas and provided helpful comments on an early draft. We also thank David Borchers and Duane Diefenbach who provided helpful comments on an early draft. The findings and conclusions in this article are those of the author(s) and do not necessarily represent the views of the U.S. Fish and Wildlife Service. Any use of trade, firm, or product names is for descriptive purposes only and does not imply endorsement by the U.S. Government.

Additional Information and Declarations

Competing Interests

Author Contributions

Animal Ethics

Field Study Permissions

Data Availability

The authors declare that they have no competing interests.

Richard J. Camp conceived and designed the experiments, performed the experiments, analyzed the data, prepared figures and/or tables, authored or reviewed drafts of the article, and approved the final draft.

David L. Miller conceived and designed the experiments, analyzed the data, authored or reviewed drafts of the article, and approved the final draft.

Stephen T. Buckland conceived and designed the experiments, authored or reviewed drafts of the article, and approved the final draft.

Steve J. Kendall conceived and designed the experiments, performed the experiments, authored or reviewed drafts of the article, and approved the final draft.

The following information was supplied relating to ethical approvals (i.e., approving body and any reference numbers):

This was an observational study thus no animals were handled in accordance with animal welfare standards of the Institutional Animal Care and Use Committee.

The following information was supplied relating to field study approvals (i.e., approving body and any reference numbers):

U.S. Fish and Wildlife, Big Island National Wildlife Refuge Complex, approved of forest bird monitoring on the Hakalau Forest Unit.

The following information was supplied regarding data availability:

Hawai‘i ‘Ākepa Loxops coccineus point-transect distance sampling data were provided by U.S. Fish and Wildlife Service, Hakalau Forest Unit of the Big Island National Wildlife Refuge Complex. The data are available from the U.S. Geological Survey: Camp (2020), Hakalau Forest National Wildlife Refuge, Hawaii Akepa point-transect survey, 2002: U.S. Geological Survey data release, https://doi.org/10.5066/P9Q9UXMZ.

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
