# Peer review of "Accounting for spatial habitat and management boundaries when estimating forest bird population distribution and density: inferences from a soap film smoother"

_PeerJ, doi:10.7717/peerj.15558_

## Round 0.1 · original submission · Major Revisions

Dear Dr. Richard J Camp,

The review of your paper is now complete; the Reviewers' reports are below. As you can see, the Reviewers present important points of criticism and a series of recommendations. We kindly ask you to consider all comments and revise the paper accordingly in order to respond fully and in detail to the Reviewers' recommendations. If this process is completed thoroughly, the paper will be accepted for a second review.

Once you have revised the paper accordingly, please submit it together with a detailed description of your response to these comments. Please, also include a separate copy of the revised paper in which you have marked the revisions made.

We very much appreciate your interest in publishing in our journal.

Yours sincerely,

Dr. Q. Wang,

Reviewer 1 ·

Basic reporting

“Inferences from a soap film smoother for estimating forest bird population distribution and density accounting for habitat and management boundaries” by Camp et al. is a clear and concise demonstration of a methodological advance for ecologists to interpolate animal densities within irregularly shaped study areas. The work is communicated well, with clear writing and effective and appropriate figures and tables.

Experimental design

The authors demonstrate the problem with using standard interpolation methods, list a variety of solutions for enforcing boundaries during interpolation, and demonstrate one method, well accepted in other disciplines, that is straightforward for ecologists to implement. As a bonus, the authors demonstrate how a spatially constant detection probability can be estimated and applied during the density interpolation process. The methods are well described, data are made available, and computing code is given in supplemental materials, making this a very useful methods paper for the community.

Validity of the findings

The authors demonstrate convincingly that the adopted interpolation method does not negatively impact accuracy or precision of global abundance estimates, while also giving a more realistic description of spatial patterning of abundance relative to standard methods. The strengths and limitations of the demonstrated approach are clearly communicated in the discussion section.

Additional comments

The manuscript in current form will make an excellent contribution to the literature on animal density estimation. I do have a minor suggestion to make the paper easier to understand during the first pass. Having the word ‘distribution’ in the title and abstract immediately put me in a Species Distribution Modeling (SDM) frame of mind. This frame of mind threw me off while trying to understand the research problem as the many covariates of a typical SDM should help to deal with habitat boundary issues. If, on the other hand, the concept of 'spatial interpolation' had been brought up immediately, then I would have understood the problem much sooner, because spatial interpolation, in its simplest form, does not rely on covariates other than spatial coordinates. Consider the title “Soap-film smoothers for spatial interpolation of animal densities across irregularly shaped study areas”.

Reviewer 2 ·

Basic reporting

In general, I found the manuscript well-written, but from my point of view, I think the authors should emphasize its main research question. In particular, I consider the introduction highly messy and chaotic, jumping from the general research framework (i.e. first, third and fifth paragraph), specific information of the study system (i.e. second and fourth paragraph), and applied and on-purpose information useful for management (i.e. sixth paragraph). In order to solve this issue I suggest that authors try to reorganize the main ideas of the present introduction in a logical way, separating the above-defined ideas. Likewise, I consider that authors spent too much space on it and that does translate into a clear message. To solve this problem, I suggest focusing and highlighting the main ideas trying to clarify and simplify the message as much as possible, or moving information to other better-fitted sections of the manuscript (i.e. Methods, Results, Discussion). In addition, Do you have any hypothesis that your methodology will improve others? What do you expect? I expect to include those insights in the very last paragraph of the introduction.

According to the methods, I consider it equally well-written but I do miss a general framework, ideally shown in a conceptual figure, providing a summary of their methodological approach. I consider the present manuscript highly methodological, meaning that it is necessary to provide information easily understandable for non-statistician researchers.

According to the results, there is information provided that its purpose is not clear to me. For instance, I do not see the point to include the thin-plate regression spline in Figure 1 as is a methodology not applied here; I think could be more fitted to an appendix; likewise, I will replace the right panel figure with the raw data provided in Figure 2. In addition, it is unclear to me the estimations along the boundary of the soap film model (L220-224); at least for me, there is no biological sense. If there is any statistical importance (i.e. the boundary is an area with a potentially higher error of model estimates), please, state this issue clearly in the manuscript. Likewise, I think that Figure 5 could be moved to the appendix, given it is not a key result; otherwise, please explain its importance.

Experimental design

From my point of view, the manuscript’s approach and methodology are relevant and rigorous, with sufficient and detailed information to be replicated in the future. However, as suggested above, I consider that the manuscript would considerably improve by clearly summarizing the methodological approach in a more visual way (see my comment above).

Likewise, I also consider it important that the authors provided clearly the main caveats of its methodology, if any. I think the authors provide a few hints about this issue in different parts of the manuscript. Still, I expect to summarize this information within a specific paragraph or section of the discussion.

Validity of the findings

I found that the author’s approach, data, and methodology are sound and of interest to animal ecologists as they help to provide new insights for predicting and mapping the density and abundance in fragmented landscapes. However, I think the authors need to strengthen and sharpen the message given in different parts of the manuscript.

·

Basic reporting

No Comment

Experimental design

This is a solid paper that builds on previous published work. Thank you for the opportunity to review it. It is an interesting paper that documents an elaboration of some earlier spatial smoothing approaches for similar data (Camp et al. 2020), using a method that explicitly models the density of birds along a known ecological boundary.

My primary question about the design of the paper is: Why was the soap-film smoother not compared to the TPRS spatial smoothing approach used in the previous published work (Camp et al. 2020, https://doi.org/10.1111/ecog.04859)? In the introduction, much of the rationale for the soap-film smoother is based on the leakage of the TPRS spatial smooths. But the soap-film smooth results were never compared to the leaky TPRS smooth. Why not? Wouldn’t this direct comparison be useful, particularly so that readers could directly assess the potential improvements over the methods used in Camp et al. 2020? For example, line 210 states “the main advantage of using the soap smooth is that there was no leakage across the pasture-forest boundary”, but leakage is a feature of the TPRS smoother and not the design-based estimates that provide the comparison in the paper.

Validity of the findings

No comment

Additional comments

My second question is about the squared-off boundary on the northern and western edges of the study area. Can you better explain why this arbitrary square boundary was necessary or useful?

My third point relates to figure 1: I have some suggestions so that it demonstrates leakage in a more compelling way and so that the demonstrated leakage is directly comparable to the output from the soap-film smooth described in the paper: 1 - the right panel should be replaced with a coloured surface which would be much easier to interpret than the contour lines - ideally, one that is directly comparable to the coloured maps in figure 3; 2 - the surface or contours in the right panel must be retransformed to the original density or count scale so the reader can interpret the values without exponentiating on the fly; 3 - one of the plots should show the locations of the observed counts (or presence and absence values), so the reader can see that the non-zero density estimates truly reflect "leakage"; and 4 - you could probably drop the "Northing" and "Easting" axis labels in the right panel, since they're already included in the bottom left panel.

I got a bit confused by the second paragraph in the discussion that introduces the concepts of “soft” and “hard” boundary conditions. These hard and soft boundary conditions are not described in the paper, in fact they’re only mentioned in this one paragraph, but given their relevance to the use and ecological interpretation of soap-film smoothers, a bit more detail might be useful. Perhaps it is as simple as explaining that it is possible to fix the boundary smooth to 0 (assuming that is actually the case), and thereby force the interior surface smooth to decline to 0 as it approaches the boundary. Then provide an example of a situation where this hard boundary approach would be relevant. Also just a thought on the terms “hard” and “soft”: I found them a bit confusing. With the soap-film smooth fit in the paper, the estimated densities at the forest-pasture edge are still subject to a hard edge beyond which densities of the bird in the pasture are assumed to be 0. So, aren’t all boundaries “hard” in this case. In fact if one was to imagine the density surface in 3 dimensions (Easting, Northing, and density on the z-axis) the “soft” boundary condition used in the paper would produce a much “harder” boundary where density could be quite high on the boundary, but then drop to 0 just outside of the boundary (i.e., a cliff at the edge of the density surface). By comparison, fixing the boundary density to 0 would force the surface to decrease smoothly towards the edge, creating a much "softer” boundary?

Finally, I have two minor suggestions that I think may help with the wording:
Line 19: delete “study are may include”, so that the last phrase of the sentence fits with the phrasing of the first. “…areas with concave arcs or that include holes of unsuitable habitat…”
Line 106: This phrasing suggests that the contrast here is between a soap film smoother and a distance sampling analysis. Since distance sampling is inherent to both approaches, perhaps just say "soap film smoother estimates to those of a non-spatial analysis" or a "design-based analysis".

---

## Round 0.2 · accepted · Accept

Dear Dr. Camp,

I am writing to inform you that your manuscript has been Accepted for publication. Congratulations!

·

Basic reporting

no comment

Experimental design

no comment

Validity of the findings

no comment

Additional comments

Thank you for replying to and addressing my comments and those of the other reviewers. I think the paper will make a valuable contribution. I have nothing else to add.